# Prevalence of Pruritus and Association with Anxiety and Depression in Patients with Nonalcoholic Fatty Liver Disease

**DOI:** 10.3390/biomedicines10020451

**Published:** 2022-02-15

**Authors:** Albrecht Boehlig, Florian Gerhardt, David Petroff, Florian van Boemmel, Thomas Berg, Valentin Blank, Thomas Karlas, Johannes Wiegand

**Affiliations:** 1Division of Hepatology, Department of Medicine II, Leipzig University Medical Center, Liebigstraße 20, 04103 Leipzig, Germany; albrecht.boehlig@medizin.uni-leipzig.de (A.B.); florian.gerhardt@medizin.uni-leipzig.de (F.G.); florian.vanboemmel@medizin.uni-leipzig.de (F.v.B.); thomas.berg@medizin.uni-leipzig.de (T.B.); 2Clinical Trial Center Leipzig, University of Leipzig, Härtelstraße 16/18, 04107 Leipzig, Germany; david.petroff.2@uni-leipzig.de; 3Integrated Research and Treatment Center Adiposity Diseases Leipzig, Faculty of Medicine, University of Leipzig, Philipp-Rosenthal-Straße 27, 04103 Leipzig, Germany; valentin.blank@medizin.uni-leipzig.de; 4Division of Gastroenterology, Department of Medicine II, Leipzig University Medical Center, Liebigstraße 20, 04103 Leipzig, Germany; thomas.karlas@medizin.uni-leipzig.de

**Keywords:** non-alcoholic steatohepatitis (NASH), pruritus, anxiety, depression, vibration controlled transient elastography (VCTE), FXR agonist

## Abstract

Patient-reported outcomes are important in nonalcoholic fatty liver disease (NAFLD). Pruritus is of special interest for evolving therapies with farnesoid X receptor (FXR) agonists. The aim of this study was to investigate the prevalence of pruritus in a real-life NAFLD cohort and analyze associations with anxiety and depression. Pruritus was assessed using a visual analogue- (VAS) and 5-D itch-scale (5-D). Anxiety and depression were evaluated by Beck’s-Depression-Inventory (BDI) and the Hospital Anxiety and Depression Scale (HADS-A, HADS-D). An optimal logistic regression model was found with a stepwise procedure to investigate variables associated with pruritus. In total, 123 NAFLD patients were recruited. VAS and 5-D were highly correlated (Spearman’s correlation coefficient 0.89). Moderate/severe pruritus was reported in 19% (VAS) and 21% (5-D) of patients. Anxiety and depression were present in 12% and 4% (HADS-A and HADS-D, respectively) and 12% (BDI) of cases. There was a significant association between VAS and BDI (*p* = 0.019). The final multivariate model for 5-D included diabetes mellitus (OR 4.51; *p* = 0.01), BDI (OR 5.98; *p* = 0.024), and HADS-A (OR 7.75; *p* = 0.011). One-fifth of NAFLD patients reported moderate or severe pruritus. 5-D was significantly associated with diabetes mellitus, depression, and anxiety. These findings should be tested in larger populations and considered in candidates for treatment with FXR agonists.

## 1. Introduction

Non-alcoholic fatty liver disease (NAFLD) is the most common chronic liver disease in the western world and has become a significant clinical and epidemic health problem. The term NAFLD encompasses a broad spectrum of diseases such as simple hepatic steatosis as well as non-alcoholic steatohepatitis (NASH), the more advanced form with a higher risk of disease progression to advanced fibrosis and liver cirrhosis [1,2,3,4].

Despite the tremendous clinical relevance, NAFLD and NASH are often considered to be clinically asymptomatic, at least if they have not progressed to cirrhosis. However, this attitude is currently changing. Prospectively controlled clinical trials for the development of drugs ameliorating NASH have assessed patient-reported outcomes (PROs) with validated and standardized questionnaires. They show significantly impaired quality of life in a substantial proportion of patients. At baseline, pruritus was reported in the placebo arm in 19%, 22%, and 15% of cases treated with the different farnesoid X receptor (FXR) agonists obeticholic acid (OCA), tropifexor, and cilofexor, respectively [5]. Pruritus is a common comorbid symptom of patients with chronic liver disease [6], which is of special interest because FXR agonists are a widely investigated and promising drug class for the treatment of NASH but are associated with drug-class-specific pruritogenic side effects that impair tolerability [5]. As an example, OCA demonstrated first that FXR agonists can improve histopathology in NASH cases; however, at the 25 mg dosage, 51% of study participants reported pruritus compared to 19% in the placebo group, and 9% vs. <1% prematurely stopped treatment due to this side effect [7]. Pruritus can be associated with other quality of life aspects, such as a history of depression, as shown in phase 3 clinical trials of the Apoptosis Signal-Regulating Kinase 1 (ASK1) inhibitor selonsertib [8]. This interaction may be important because depression is independently associated with NAFLD [9,10].

Note, however, that NASH patients in biopsy-controlled clinical trials are highly selected and may not be representative of real-world populations. At our center, fibrosis stages F2/F3 and F4 along with a NAFLD activity score (NAS) ≥4 points, which are central inclusion criteria in many clinical trials, was found in only 1.7% and 0.5% of cases, respectively [11]. Thus, it is of importance to analyze patient-reported outcomes not only in clinical studies but also in less-selected NASH populations, to generate evidence of the patient’s quality of life and identify variables that may impair the use of FXR agonists outside of clinical trials. Real-life data on pruritus are scarce so far. One Japanese group reported a pruritus prevalence of 45% in NAFLD patients [12], but to the best of our knowledge, pruritus data from Caucasian populations outside of clinical trials have not been reported yet. Thus, we investigated the prevalence of pruritus in prospectively recruited NAFLD patients and analyzed associations with anxiety and depression.

## 2. Materials and Methods

### 2.1. Ethical Statement

This study was performed according to the guidelines for good clinical practice (E6/R1) and the ethical guidelines of the Helsinki Declaration and was approved by the local ethical committee (University of Leipzig, ethical vote 261/19-ek). Written informed consent was obtained from all participants. The study was registered in the German Clinical Trials Register (DRKS00017767).

### 2.2. Study Population

From June 2019 to December 2020, adult patients (≥18 years) with a clinical diagnosis of NAFLD were prospectively recruited at the outpatient unit of our tertiary referral center. The target population comprised outpatients with hepatic steatosis detected by imaging methods, irrespective of their fibrosis stage. Participants with compensated cirrhosis (Child-Pugh A) were also eligible for inclusion, while individuals with Child-Pugh B or C cirrhosis were excluded. Patients with known psychiatric comorbidities such as depression or anxiety disorders, dermatologic comorbidities such as psoriasis vulgaris, or patients after solid organ transplantation were excluded from participation in the study. All participants were not currently enrolled in another clinical trial for NAFLD therapy. Pregnant or breastfeeding women were not allowed to participate. Significant alcohol consumption defined as >30 g per day for men and >20 g per day for women as well as the presence of chronic liver diseases other than NAFLD, such as viral hepatitis, was another exclusion criterion [13]. In addition, patients with hepatocellular carcinoma and other active malignancies as well as benign liver tumors, for example, hepatocellular adenoma, were excluded.

All patients were thoroughly evaluated: Comorbidities including type 2 diabetes mellitus and medication were recorded using a standardized questionnaire. In addition, anthropometric data (weight, height, body mass index) were collected. Results of liver function tests were extracted from the same visit.

### 2.3. Pruritus, Anxiety, and Depression

Pruritus was assessed using a visual analogue scale (VAS, see Figure 1) and the 5-D itch scale (5-D) [14,15]. The five different components of the 5-D itch scale are listed in Table 1. Pruritus according to the two scales was classified (VAS/5-D) as absent (0/5–8), mild (1–3/9–11), moderate (4–6/12–17), severe (7–8/18–21), or very severe (9–10/22–25), see Table 2. All pruritus questionnaires referred to the timespan of the past two weeks.

Depression was evaluated using Beck’s Depression-Inventory (BDI) questionnaire. Its overall sum score is classified as normal (0 to 10 points), borderline (11 to 17 points), or clinically relevant (18 to 62 points) [16].

The Hospital Anxiety and Depression Scale (HADS-A for anxiety and HADS-D for depression) was used as a second questionnaire for the evaluation of anxiety and depression. The overall sum score is calculated separately for HADS-A and HADS-D and can be classified as normal (0 to 7 points), borderline (8 to 10 points), or clinically relevant (11 to 21 points). For further details see Bjelland et al. and Deterding et al. [17,18].

### 2.4. Fibrosis and Steatosis

The severity of NAFLD defined by estimates of fibrosis and steatosis was assessed with vibration-controlled transient elastography (VCTE) including liver stiffness measurement (LSM) and a controlled attenuation parameter (CAP) as described before [19]. The FibroScan-AST score (FAST) based on LSM, CAP, and aspartate-aminotransferase (AST) was calculated to identify patients with active NASH (NAS-score ≥ 4) and significant fibrosis (F ≥ 2) [20].

Laboratory-based non-invasive fibrosis scores were calculated using the NAFLD fibrosis (NFS) and the fibrosis 4 (FIB-4) scores [21,22]).

### 2.5. Statistical Analysis

Statistical analyses were performed with R (Version 4.0.4). Means and standard deviations are denoted by X ± Y and medians and interquartile range by X [Y, Z].

A step-wise procedure was used with forward and backward selection based on Akaike’s information criterion to select an optimal logistic regression model to investigate variables associated with pruritus. The variables considered were sex, age, BMI, diabetes, hypertension, hypercholesterolemia, CAP, LSM (on a logarithmic scale), FIB-4 (on a logarithmic scale), FAST, and the categorical variables for BDI HADS-A and HADS-D, which were all treated on a binary scale (relevant or not) due to the small sample size. NFS was not considered due to missing values and collinearity with LSM and FIB-4. Contingency tables were analyzed with Fisher’s Exact Test, and the confidence interval for Spearman’s correlation coefficient was found using bootstrapping [23].

## 3. Results

### 3.1. Population Characteristics

In total, 137 NAFLD patients were prospectively screened for the study. Fourteen participants dropped out as screening failures, mostly because of relevant comorbidities and language barriers, leaving 123 NAFLD patients, who were included in the preliminary analysis. Two patients were excluded from the cohort due to missing VAS data, leaving 121 patients for the final analysis. Baseline characteristics of the population are listed in Table 3: 53% were female, the mean age was 56 years, and the mean BMI was 30 kg/m^2^. Furthermore, 12% of patients had liver cirrhosis. Although chronic kidney disease was not an exclusion criterion, only one participant had advanced chronic kidney disease (glomerular filtration rate (GFR) < 30 mL/min/1.73 m^2^). Moreover, 119 of the 121 patients (98%) in our cohort had normal bilirubin levels, while 2 cases presented with bilirubin 1.67 and 2.08 times the upper limit of normal (ULN). There were no significant differences in age, sex, or BMI, nor were there significant differences in CAP, LSM, and the non-invasive fibrosis scores between the patients with no or mild pruritus compared to individuals with moderate or severe pruritus.

### 3.2. Pruritus

The distributions of VAS and 5-D scores are shown in Table 4. The 5-D itch score was not as easy to utilize as VAS, resulting in unavailable 5-D itch scores from 11 patients. These unavailable scores were the result of incomplete questionnaire responses by the participants. According to VAS, no/mild/moderate/severe pruritus was observed in 59/39/18/5 patients, respectively. Using the 5-D itch scale, no/mild/moderate/severe pruritus was reported by 64/23/20/3 patients, respectively.

The patient with GFR 29 mL/min/1.73 m^2^ presented with mild pruritus according to VAS and 5-D itch scores, the individual with bilirubin 1.67× ULN reported mild (VAS)–moderate (5-D) pruritus, and the case with bilirubin 2.08× ULN had no pruritus according to VAS and 5-D itch scores.

Very severe pruritus was not reported by either the VAS or 5-D assessment. In summary, clinically relevant pruritus, that is, moderate or severe pruritus, was observed in 19% (VAS) and 21% (5-D) of patients.

The mean VAS score was 1.7 ± 2.2 and the mean 5-D itch-score was 8.3 ± 4.0, which is illustrated in Figure 2. There was a strong correlation between the two parameters (Spearman 0.89, 95% CI 0.83 to 0.93).

### 3.3. Anxiety and Depression

The distribution of BDI, as well as HADS-A and HADS-D scores, is presented in Table 5. Due to incomplete questionnaire responses, BDI data were available from 111 and HADS-A/HADS-D data from 119 participants. According to BDI, a relevant score was observed in 13 (12%) of participants. Using the HADS-D questionnaire, five (4%) patients had a relevant score. Clinically relevant anxiety (HADS-A > 10) was present in 14 (12%) individuals.

The patient with GFR 29 mL/min/1.73 m^2^ presented with normal results in the BDI, HADS-A, and HADS-D assessments, while the individual with bilirubin 1.67× ULN reported a relevant HADS-A result of 12 points and borderline results of BDI and HADS-D. The patient with bilirubin 2.08× ULN had normal results in the BDI, HADS-A, and HADS-D assessments.

### 3.4. Associations between Pruritus and Covariables

The associations between VAS and BDI as well as HADS scores are listed in Table 6. There was a significant association between VAS and BDI (*p* = 0.019). The associations between VAS and HADS-A (*p* = 0.18), as well as VAS and HADS-D (*p* = 0.22), were not statistically significant.

The final multivariate model for VAS included CAP with odds ratios (OR) for moderate/severe itching per 10 dB/m of 1.06 (95% CI 0.97–1.17; *p* = 0.17) and relevant HADS-A (OR 2.78, 95% CI 0.75-9.53; *p* = 0.12). Other variables were not statistically relevant in this model.

For the 5-D scale, the analogous model included type 2 diabetes mellitus (T2DM; OR 4.51; 95% CI 1.43-15.3; *p* = 0.012), relevant BDI (OR 5.98; 95% CI 1.27–28.8; *p* = 0.022), and relevant HADS-A (OR 7.75; 95% CI 1.62–40.1; *p* = 0.011). As a sensitivity analysis, a bootstrap method with 10,000 repetitions estimated odds ratios of 4.87 (1.46–20.9), 6.79 (0.90–57.9), and 8.73 (1–13–123) for the three covariables T2DM, BDI, and HADS-A, respectively.

When using the same model with VAS, these associations were not statistically significant (T2DM: 1.66 (0.58–4.63), *p* = 0.34; BDI: 2.39 (0.56–9.18), *p* = 0.23; HADS-A: 2.67 (0.63–10.4), *p* = 0.17).

## 4. Discussion

Patient-reported outcomes are increasingly important in NAFLD, and pruritus is of special interest for evolving therapies with FXR agonists. Clinical trials with this drug class have started to investigate pruritus in controlled prospective settings and described a prevalence of 4-19% in placebo-treated individuals [7,24,25,26].

The primary aim of our study was to analyze the prevalence of pruritus in individuals with NAFLD in a real-life approach because NAFLD patients in clinical trials are highly selected [11,27] and clinical real-life data are scarce so far. One Japanese study observed a pruritus prevalence of 45% in 338 NAFLD patients [12], and a second study reported pruritus in 2/14 (14%) cases [28]. In our NAFLD population, moderate and severe pruritus was present in one-fifth of individuals. If the overall intensity of pruritus is considered, the prevalence was 51% according to VAS and 42% according to the 5-D itch scale. A comparison of pruritus in individuals with liver disease to the general population seems difficult as many highly prevalent diseases such as diabetes mellitus, chronic kidney disease, or skin diseases, which can also cause pruritus, would have to be excluded to create an adequate control group. Other representative studies on individuals of the German working population and the German general population found the prevalence of pruritus to be approximately 7 to 17% [29,30,31]. Therefore, we found the prevalence of pruritus to be higher in our NAFLD cohort compared to individuals without liver disease. Our results are the first real-life data in Caucasian patients and seem to be comparable to the Oeda study. Pruritus was assessed with two independent and validated methods, which were highly correlated with each other. The 5-D itch scale characterizes pruritus in more detail than the VAS; however, it might be more complex to use in a real-life setting, because results were only available in 110 patients compared to 121 cases in whom the VAS could be interpreted correctly.

The prevalence of pruritus in controlled clinical trials with obeticholic acid, cilofexor, and EDP-305 was 19%, 4%, 15%, and 4% at baseline in placebo-treated individuals, respectively [7,24,25,26]; however, these data are difficult to compare with the Japanese data and our real-life data due to the study-specific selection of patients. As pruritus seems to be a major symptom in individuals with NAFLD, standardized evaluation prior to therapy with FXR agonists should be considered.

The secondary aim of our study was to identify clinical factors that may be associated with pruritus in cases with NAFLD. For pruritus measured by the 5-D itch scale, we observed a significant association with diabetes mellitus, depression according to BDI, and anxiety (HADS-A), whereas similar results were not obtained with the VAS pruritus assessment. The association with the 5-D results should not be over-interpreted, because the wide confidence intervals indicate that they are prone to a type 1 error due to the limited sample size. In the real-life study by Oeda et al., a significant association between pruritus and diabetes mellitus was described, but this analysis was carried out in chronic liver diseases of different etiologies and not restricted to NAFLD patients [12]. In the setting of controlled trials in patients with pre-cirrhosis and cirrhosis (F3 and F4 fibrosis), Younossi et al. reported an association between pruritus and the female gender, lower serum albumin, depression, nervous system, and dermatologic comorbidities [8]. In their study, pruritus was measured with the Chronic Liver Disease Questionnaire–NASH (CLDQ-NASH). In addition to the differences between real-life and clinical studies, future studies should consider whether different diagnostic tools such as VAS, the 5-D itch scale, or CLDQ-NASH reveal similar interactions between pruritus and concomitant diseases. With respect to concomitant diseases, diabetes mellitus, anxiety, and depression will be of special interest: (i) Diabetes mellitus is one of the major concomitant diseases in NAFLD patients (33% in the present cohort), patients with diabetes mellitus are known to have a high rate of pruritus, which is often associated with xerosis cutis [32,33,34], and diabetes mellitus was associated with pruritus in the Oeda study. (ii) Secondly, anxiety and depression are independently associated with NAFLD and may therefore also interact with pruritus [8,9,10]. Depression and anxiety may even be the primary cause of pruritus, especially in patients with high pruritus intensity or impaired quality of life [35,36,37]. These patients tend to report more pruritus triggers and use a higher number of emotional adjectives to describe their pruritus. If this is observed in individual cases, one should consider involving a psychologist or psychiatrist in diagnosis and therapy. It is known that anxiety and depression have the potential to be pruritus-increasing factors, so psychological interventions should also be considered in the treatment of pruritus in general [35].

Subgroup analyses on the prevalence of pruritus in patients with diabetes mellitus, anxiety, or depression and the potential impact on the tolerability of FXR agonists have not been performed in the clinical NASH trials published so far [7,24,25,26]. Hence, drug therapy with FXR agonists should be critically evaluated in NAFLD patients who already complain about clinically significant pruritus prior to treatment initiation. In light of this, it should also be investigated whether the use of non-bile acid FXR agonists, such as cilofexor, might be more suitable for the therapy of NAFLD patients with pruritus [38].

Our study results are limited by the relatively small sample size and the monocentric approach. Especially concerning the interaction between anxiety and depression with pruritus and the fact that overall impaired quality of life might hint at a psychological etiology of the pruritus, it would have been interesting to apply a quality-of-life questionnaire such as the Public Health Questionnaire (PHQ-9), which, however, was not incorporated in the design of our study.

In conclusion, one-fifth of Caucasian NAFLD patients reported moderate or severe pruritus in this real-life setting. The 5-D itch scale and the VAS were highly correlated. Diabetes mellitus, depression, and anxiety may be associated with pruritus and might affect many aspects of the patient’s daily life and overall wellbeing. Therefore, these associations should be tested in larger populations and carefully considered in candidates for treatment with FXR agonists.

## Figures and Tables

**Figure 1 biomedicines-10-00451-f001:**
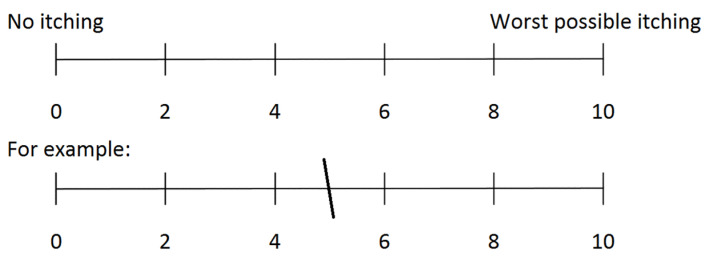
Visual Analogue Scale for the assessment of pruritus. The subjects should draw a line anywhere on the given scale that best represents the severity of their itching.

**Figure 2 biomedicines-10-00451-f002:**
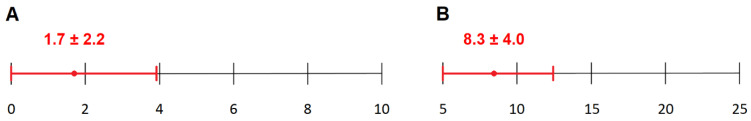
(**A**) Visual analogue scale with mean ± SD. (**B**) 5-D itch scale with mean ± SD (indicated in red font).

**Table 1 biomedicines-10-00451-t001:** Components of the 5-D itch scale. All of these components are to be rated separately during the timespan of the past two weeks. The scores for each component are summed to an overall score ranging from a minimum of 5 to a maximum of 25 points [14].

Component	Features
Duration	Hours of itching per day
Degree	Intensity of itching
Direction	Change of ichting intensity
Disability	Impact of itching on activities such as sleep, leisure/social, housework/errands, work/school
Distribution	Number of affected body parts

**Table 2 biomedicines-10-00451-t002:** Definition of the ordinal pruritus variables from the visual analogue scale and 5-D itch scale. VAS: Visual analogue scale, 5-D: 5-D itch scale.

Intensity of Pruritus	VAS	5-D
Absent (No)	0	5–8
Mild	1–3	9–11
Moderate	4–6	12–17
Severe	7–8	18–21
Very severe	9–10	22–25

**Table 3 biomedicines-10-00451-t003:** Population characteristics for all patients, dichotomized according to pruritus severity assessed by VAS. Values are frequency (%), mean ± SD, or median [IQR]. BMI: Body mass index, CAP: Controlled attenuation parameter, FIB-4: Fibrosis 4 score, GGT: gamma-glutamyl transferase, NFS: NAFLD fibrosis score.

	All Patients (*n* = 121)	No or Mild Pruritus (*n* = 98)	Moderate or Severe Pruritus (*n* = 23)	*p*-Value
Female	64 (53%)	50 (51%)	14 (61%)	0.39
Age [years]	55.8 ± 14.5	55.9 ± 14.9	55.5 ± 13.3	0.92
BMI [kg/m^2^]	29.8 ± 5.1	29.7 ± 4.9	29.8 ± 6.1	0.94
<25	15 (12%)	9 (9%)	6 (26%)	
25–30	55 (45%)	49 (50%)	6 (26%)	
30–35	32 (26%)	24 (24%)	8 (35%)	
35–40	14 (12%)	12 (12%)	2 (9%)	
>40	5 (4%)	4 (4%)	1 (4%)	
Presence of cirrhosis on abdominal ultrasound	14 (12%)	11 (11%)	3 (13%)	0.82
Diabetes mellitus	40 (33%)	31 (32%)	9 (39%)	0.49
Hypertension	74 (61%)	63 (64%)	11 (48%)	0.14
Hypercholesterolemia	46 (38%)	36 (37%)	10 (43%)	0.55
CAP [dB/m]	315 ± 54	312 ± 55	330 ± 48	0.16
LSM [kPa]	5.8 [4.3, 7.0]	5.9 [4.3, 7.0]	4.8 [4.3, 9.2]	0.69
NFS	−1.70 [−3.11, −0.83]	−1.75 [−3.11, −0.83]	−1.70 [−3.11, −0.30]	0.57
FIB-4	1.18 [0.82, 1.98]	1.17 [0.82, 1.98]	1.19 [0.82, 1.94]	0.75
FAST Score	0.32 [0.19, 0.51]	0.35 [0.19, 0.51]	0.28 [0.19, 0.55]	0.79
ALT [µkat/L]				
female	0.60 [0.41, 0.66]	0.64 [0.41, 0.66]	0.56 [0.41, 0.83]	0.71
male	0.86 [0.59, 1.15]	0.85 [0.59, 1.15]	1.02 [0.59, 1.09]	0.40
AST [µkat/L]				
female	0.54 [0.42, 0.70]	0.56 [0.42, 0.70]	0.46 [0.42, 0.79]	0.61
male	0.60 [0.46, 0.69]	0.59 [0.46, 0.69]	0.61 [0.46, 0.76]	0.89
GGT [µkat/L]	1.06 [0.61, 2.17]	1.06 [0.61, 2.17]	0.90 [0.61, 2.04]	0.96
GFR [mL/min/1.73m^2^]	86.0 ± 20.2	87.0 ± 20.0	82.0 ± 20.9	0.29

**Table 4 biomedicines-10-00451-t004:** Distribution of the VAS (A) and 5-D itch scale (B) results. Values are frequencies.

**A**											
VAS	0	1	2	3	4	5	6	7	8	9	10
N	59	17	8	14	6	6	6	4	1	0	0
Pruritus intensity	No	Mild			Moderate			Severe		Very severe	
**B**											
5-D	5–8	9–11			12–17			18–21		22–25	
N	64	23			20			3		0	
Pruritus intensity	No	Mild			Moderate			Severe		Very severe	

**Table 5 biomedicines-10-00451-t005:** Distribution of BDI as well as HADS-A and HADS-D scores in the population. Values are frequencies (%).

Score Classification	Normal Score	Borderline Score	Relevant Score
BDI	88 (79%)	10 (9%)	13 (12%)
HADS-A	86 (72%)	19 (16%)	14 (12%)
HADS-D	99 (83%)	15 (13%)	5 (4%)

**Table 6 biomedicines-10-00451-t006:** Contingency tables for the associations between VAS/BDI (A, *p* = 0.019), VAS/HADS-A (B, *p* = 0.18), and VAS/HADS-D (C, *p* = 0.22); *p*-values are based on Fisher’s exact test.

**A**		BDI		
		Normal Score	Borderline Score	Relevant Score
VAS	None	48	2	4
	Mild	27	4	4
	Moderate	12	2	4
	Severe	1	2	1
**B**		HADS-A		
		Normal score	Borderline score	Relevant score
VAS	None	44	9	5
	Mild	28	6	4
	Moderate	12	4	2
	Severe	2	0	3
**C**		HADS-D		
		Normal score	Borderline score	Relevant score
VAS	None	52	5	1
	Mild	30	6	2
	Moderate	14	3	1
	Severe	3	1	1

## Data Availability

Data can be shared upon individual request.

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
