# Peer review of "Prevalence of Pruritus and Association with Anxiety and Depression in Patients with Nonalcoholic Fatty Liver Disease"

_biomedicines, 2022, doi:10.3390/biomedicines10020451_

Round 1

Reviewer 1 Report

Dear Authors,

The manuscript entitled "Prevalence of pruritus and association with anxiety and depression in patients with nonalcoholic fatty liver disease” is an interesting topic. It is very well structured but could be better reasoned.

I have a few suggestions.

1. Rethink the keywords, the ones that make most sense.

I suggest the inclusion of “pruritus”, “anxiety”, “depression”, and the exclusion of some that did not make so much sense.

2. Do not repeat definitions of abbreviations.

(line 45 “obeticholic acid (OCA)” and line 49

3. Define abbreviations:

Line 54 - ”ASK1”

Line 135 – “AST”

4. Line 114 – Figure 1 (VAS) - the size of the picture can be reduced

5. Line 160 – Rewrite “30 kg/m2. 12%”

6. Line 218 – T2DM should be referred to here and excluded from the keywords

7. “Pruritus can be associated with other quality of life aspects like a history of depression,”

It would also be interesting to understand if pruritus can't trigger anxiety and depression? Is there anything in the literature?

8. The discussion should be improved.

The focus of the study should not be so much the comparison with patients with diabetes but with anxiety and depression. How is pruritus affected by these conditions (anxiety and depression) and how do these conditions affect pruritus...

9. How interesting would it have been to use a quality of life assessment questionnaire? For example Public Health Questionnaire (PHQ-9).

10. The limitations of the study should also be highlighted.

Author Response

Dear Prof. Mousa and Editorial Board,

Thank you very much for giving us the opportunity to revise our manuscript entitled “Prevalence of pruritus and association with anxiety and depression in patients with nonalcoholic fatty liver disease”. We believe the manuscript has benefited from the changes made and provide a point-to-point response to your review comments below. Changes in the manuscript are highlighted.

Best regards

Albrecht Boehlig and Johannes Wiegand

Reviewer #1

  1. Rethink the keywords, the ones that make most sense. I suggest the inclusion of “pruritus”, “anxiety”, “depression”, and the exclusion of some that did not make so much sense.

    We reorganized the keywords and included “pruritus”, “anxiety” and “depression” and deleted “type 2 diabetes mellitus”.
  2. Do not repeat definitions of abbreviations. (line 45 “obeticholic acid (OCA)” and line 49)

    We changed “obeticholic acid (OCA) in line 49 in order not to repeat defined abbreviations.
  3. Define abbreviations: Line 54 - ”ASK1”, Line 135 – “AST”

    We defined the abbreviations “ASK1” (line 54) and “AST” (line 135) as suggested.
  4. Line 114 – Figure 1 (VAS) - the size of the picture can be reduced

    The size of the picture in Figure 1 has been reduced.
  5. Line 160 – Rewrite “30 kg/m2. 12%”

    The term “30 kg/m2. 12%” has been rewritten correctly.
  6. Line 218 – T2DM should be referred to here and excluded from the keywords

    In line 228, we changed “T2DM” to “type 2 diabetes mellitus” and, as mentioned in No. 1, deleted it from the keywords.

7./8.      “Pruritus can be associated with other quality of life aspects like a history of depression,” It would also be interesting to understand if pruritus can't trigger anxiety and depression? Is there anything in the literature? The discussion should be improved. The focus of the study should not be so much the comparison with patients with diabetes but with anxiety and depression. How is pruritus affected by these conditions (anxiety and depression) and how do these conditions affect pruritus...

We added more thoughts and literature to the discussion illustrating the psychological contributing factors to pruritus and tried to focus the discussion more on associations between pruritus and anxiety/depression than on diabetes mellitus.

9./10.    How interesting would it have been to use a quality of life assessment questionnaire? For example Public Health Questionnaire (PHQ-9). The limitations of the study should also be highlighted.

We added the limitations of our study and added a comment about the Public health Questionnaire.

Reviewer 2 Report

Title: Prevalence of pruritus and association with anxiety and depression in patients with nonalcoholic fatty liver disease.

Authors: Albrecht Boehlig, Florian Gerhardt, David Petroff, Florian van Boemmel, Thomas Berg, Valentin Blank, Thomas Karlas, Johannes Wiegand.

General comment:

Pruritus is one of the significant limitations in using some farnesoid X receptor (FXR) agonists in NAFLD treatment. Therefore it is critical to identify patients at a high risk of this side effect – e.g., those who suffer from pruritus spontaneously. In their work, Albrecht Boehlig et al. assessed the prevalence of pruritus and its association with anxiety and depression in a real-life cohort of NAFLD patients. They found that approximately 20% of NAFLD patients report moderate or severe pruritus. In these individuals, the score of the 5-D itch scale is significantly associated with diabetes mellitus, depression, and anxiety. The concept of the manuscript is clear, and the results might be significant for everyday clinical practice. However, the authors should explain some methodological issues before the manuscript is accepted for publication.

Major revisions:

Abstract:

  • The study's aims could be formulated more clearly.

Material and methods:

  • In the Discussion, the authors write: “A comparison of pruritus in individuals with liver disease to the general population seems difficult as many high-prevalent diseases such as diabetes mellitus, chronic kidney disease or skin diseases which can also cause pruritus would have to be excluded to create an adequate control group.” – it is not clear from the study group description if individuals with chronic kidney disease were excluded from the study protocol.
  • Considering the possible confounders influencing pruritus intensity, the authors should also assess the bilirubin level.

Results:

  • Please explain the reasons for the incomplete data collection:

Table 4 – results of VAS were obtained from 121 study participants, while 5-D – from 110.

Table 5 – BDI were obtained in 111 study participants, while HADS-D and HASD-A in 119

It is not clear from the study protocol that not all patients completed the questionaries mentioned above.

Minor revisions:

Abstract:

  • “Anxiety and depression were present in 12% and 4% (HADS-D) and 12% (BDI) of cases." – this sentence is slightly unclear – please consider changing it to, e.g., "Anxiety and depression were present in 12% and 4% (HADS-A and HADS-D, respectively) and 12% (BDI) of cases”.

Whole manuscript:

  • Please develop the abbreviations as only they occur in the text. E.g., ASK1, AST, T2DM.

Author Response

Reviewer #2

General comment: Pruritus is one of the significant limitations in using some farnesoid X receptor (FXR) agonists in NAFLD treatment. Therefore it is critical to identify patients at a high risk of this side effect – e.g., those who suffer from pruritus spontaneously. In their work, Albrecht Boehlig et al. assessed the prevalence of pruritus and its association with anxiety and depression in a real-life cohort of NAFLD patients. They found that approximately 20% of NAFLD patients report moderate or severe pruritus. In these individuals, the score of the 5-D itch scale is significantly associated with diabetes mellitus, depression, and anxiety. The concept of the manuscript is clear, and the results might be significant for everyday clinical practice. However, the authors should explain some methodological issues before the manuscript is accepted for publication.

Thank you for this overall positive assessment. We provide more details on methodological issues as suggested below.

Whole manuscript: Please develop the abbreviations as only they occur in the text. E.g., ASK1, AST, T2DM.

We defined all abbreviations in the text, such as ASK1, AST, T2DM.

Abstract:  The study's aims could be formulated more clearly.

We rewrote the aims of the study to be more clear.

Abstract: “Anxiety and depression were present in 12% and 4% (HADS-D) and 12% (BDI) of cases." – this sentence is slightly unclear – please consider changing it to, e.g., "Anxiety and depression were present in 12% and 4% (HADS-A and HADS-D, respectively) and 12% (BDI) of cases”.

We changed the sentence as suggested.

Material and methods: In the Discussion, the authors write: “A comparison of pruritus in individuals with liver disease to the general population seems difficult as many high-prevalent diseases such as diabetes mellitus, chronic kidney disease or skin diseases which can also cause pruritus would have to be excluded to create an adequate control group.” – it is not clear from the study group description if individuals with chronic kidney disease were excluded from the study protocol. Considering the possible confounders influencing pruritus intensity, the authors should also assess the bilirubin level.

We added information about bilirubin levels and GFR of the participants and found only three patients with values outside the normal range, rendering conclusive statements impossible. We now provide details on these patients in the manuscript.

Results: Please explain the reasons for the incomplete data collection. Table 4 – results of VAS were obtained from 121 study participants, while 5-D – from 110. Table 5 – BDI were obtained in 111 study participants, while HADS-D and HASD-A in 119. It is not clear from the study protocol that not all patients completed the questionnaires mentioned above.

We added information to explain the reasons for incomplete data collection regarding the BDI as well as HADS-A and HADS-D.

Round 2

Reviewer 2 Report

I want to express my gratitude for the opportunity to re-review the paper entitled: "Prevalence of pruritus and association with anxiety and depression in patients with nonalcoholic fatty liver disease" by Albrecht Boehlig et al. Since the authors addressed my concerns regarding the methodology and presentation of results, I find the manuscript acceptable for publication in Biomedicines.